# Microfluidics: Insights into Intestinal Microorganisms

**DOI:** 10.3390/microorganisms11051134

**Published:** 2023-04-27

**Authors:** Ping Qi, Jin Lv, Xiangdong Yan, Liuhui Bai, Lei Zhang

**Affiliations:** 1The First Clinical Medical College, Lanzhou University, Lanzhou 730000, China; 2Department of General Surgery, The First Hospital of Lanzhou University, Lanzhou 730000, China; 3Key Laboratory of Biotherapy and Regenerative Medicine of Gansu Province, The First Hospital of Lanzhou University, Lanzhou 730000, China

**Keywords:** microfluidics, intestinal microorganisms, intestine-on-a-chip, microfluidic drug delivery system, droplet microfluid, electrospray microfluid

## Abstract

Microfluidics is a system involving the treatment or manipulation of microscale (10^−9^ to 10^−18^ L) fluids using microchannels (10 to 100 μm) contained on a microfluidic chip. Among the different methodologies used to study intestinal microorganisms, new methods based on microfluidic technology have been receiving increasing attention in recent years. The intestinal tracts of animals are populated by a vast array of microorganisms that have been established to play diverse functional roles beneficial to host physiology. This review is the first comprehensive coverage of the application of microfluidics technology in intestinal microbial research. In this review, we present a brief history of microfluidics technology and describe its applications in gut microbiome research, with a specific emphasis on the microfluidic technology-based intestine-on-a-chip, and also discuss the advantages and application prospects of microfluidic drug delivery systems in intestinal microbial research.

## 1. Introduction

The intestinal tract is not only the largest digestive organ in the human body, but also the habitat of a diverse population of microorganisms. The intestinal microflora include symbiotic microorganisms that colonize the intestinal tract and grow synergistically with epithelial cells [1]. This diverse community, comprising bacteria, fungi, viruses, and bacteriophages, is essential to the maintenance of intestinal homeostasis [2]. Human microbiota gene sequencing studies have revealed that an imbalance in the intestinal microbiota is associated with a range of diseases, including inflammatory bowel disease, metabolic syndrome, non-alcoholic fatty liver disease, and colorectal cancer [3,4,5,6]. However, current knowledge regarding these associations is relatively rudimentary and warrants further extensive investigation [7].

In recent years, the development of microfluidic technology has presented an array of new opportunities for studying the gut microbiome in a more precise and controlled manner. In this context, the intestine-on-a-chip approach provides a convenient platform that can applied to reconstruct the interface between the intestinal lumen and capillary tissues, thereby simulating, in vitro, the in vivo microenvironment (oxygen gradient, oxygen partial pressure, carbon dioxide concentration, and immune components) [8]. In these chips, several channels can be used for co-culturing microvascular endothelial cells, immune cells, and commensal and pathogenic microorganisms obtained from the human body. Thus, by applying microfluidic chips to construct on-chip gut models, microfluidic technology can more accurately simulate the in vivo microenvironment and investigate the interactions between microorganisms and the gut environment. In addition, these chips can be used to fabricate microcapsules to develop microfluidic drug delivery systems [9]. These chips utilize alginate, chitosan, whey protein, and similar biocompatible substances to encapsulate microorganisms within microcapsules, thereby protecting living microorganisms from being directly exposed to the external environment, and also enabling their use in the human body [10,11]. Such microfluidic drug delivery systems can be used to deliver active microorganisms to the intestine, and thereby facilitate investigations assessing the effects of gut microbiota on the host.

Microfluidic technology accordingly has the advantages of simulating the actual physiological environment of the host gut, co-cultivating multiple microorganisms, co-cultivating microorganisms with host cells, and delivering active microorganisms, thereby emphasizing the potential utility of this technology for the study of the gut microbiota and host–microbiota interactions. Microfluidic technology thus provides a powerful tool for in-depth research on the structure and function of the gut microbiota, and its relevance to host health.

## 2. The Development of Microfluidics

Microfluidic technology was first proposed in 1990 by Manz et al. [12], and was implemented the following year on a flat microchip to achieve capillary electrophoresis. In 1994, Jacobson et al. improved the injection method, thereby enhancing the performance and practicability of microfluidic chip capillary electrophoresis [13]. In 1995, Mathies et al. established high-speed DNA sequencing on chips using microfluidic technology [14]. In 1996, polymerase chain reaction (PCR) and capillary electrophoresis were integrated on microfluidic chips [15], and in the following year, multi-channel capillary electrophoresis DNA sequencing was achieved on microfluidic chips [16]. In 2002, in a report titled “Microfluidic large-scale integrated chip” published in magazine, Thorsen et al. introduced microfluidic chips integrated with thousands of microvalves and hundreds of microreactors [17], marking a fundamental leap from simple electrophoresis technology to elaborate multifunctional integrated laboratories. The history of microfluidic chip development is summarized in Table 1.

With ongoing advances in material science, the manufacture of microfluidic chips has become increasingly complex, and their applications are becoming more extensive and interdisciplinary. For example, scientists have performed short-term analyses of cells on microfluidic chips and demonstrated the migration and functions of cells in microchannels [24]. To date, viable cell culture chips have been designed with a controllable microenvironment that can simulate the functions and characteristics of living organs. Microfluidic chips provide a powerful tool to mimic human intestinal physiological and pathological conditions, and can be used to assess the associations of these condition with the gut microbiota, as well as providing a platform for the development and evaluation of drugs [25]. In this latter regard, advances in droplet and electrospray microfluidic technology have enabled the development of microfluidic drug delivery systems that can be used to prepare drug-loaded microcapsules that protect the active components and facilitate the controlled release of drugs and other active substances in vivo [26,27,28]. Figure 1A shows a selection of the different types of microfluidic chips designed to date.

## 3. Application of Microfluidic Intestine-on-a-Chip

Microfluidic intestinal chips create a 3D model of the living intestine established in the chip cavity to accurately reflect the physiological and pathological status of the human body. For example, chips can be designed to simulate the biomechanical characteristics of intestinal peristalsis and fluid flow, thereby enabling intestinal epithelial cells cultured on the chip to differentiate normally and form a miniature intestinal villous structure. Importantly, using these chips, epithelial cells and intestinal microorganisms can be co-cultured for varying lengths of time, from several days to several weeks, thereby facilitating long-term experimentation [30,32]. The process of microfluidic intestinal chip construction is illustrated in Figure 1B. Intestine-on-a-chip systems can simulate the physiological microenvironment of the intestine for culturing gut microorganisms, and in this regard, Kasendra et al. have created a system that combines the epithelial and microvascular endothelial cells obtained from intestinal biopsy samples and simulated the liquid flow and peristaltic movement of the intestinal lumen [31]. The gut-on-a-chip shown in Figure 1C, which is designed to simulate the liquid flow and peristaltic motion in the intestinal lumen, comprises a flexible transparent PDMS support layer with upper and lower microfluidic channels and left and right lateral chambers separated by a porous PDMS membrane coated with an extracellular matrix. The stretchable nature of PDMS enables the chip to adjust to the negative pressure applied to the side chambers in response to cyclic suction applied to the vacuum side via tubing. This cyclic suction causes the central porous PDMS membrane to stretch and the monolayer of Caco-2 cells on the membrane to undergo mechanical deformation, thereby mimicking the physiological peristaltic motion of the intestinal lumen at a frequency of 0.15 Hz. Additionally, the chip can simulate the flow of fluid in the intestinal lumen by controlling the low-rate (30 μL h^−1^) liquid flow in the central microchannel, which generates a low shear stress of 0.02 dyne cm^−2^ [33]. Figure 1D presents a schematic depiction of the process of creating microfluidic co-cultures between primary human intestinal epithelium and intestinal microvascular endothelium using the Intestine Chip. Cells cultured on such chips can differentiate into four types of intestinal cells, namely, absorptive, mucus-secreting, intestinal endocrine, and Panetta cells, which spontaneously undergo development to yield intestinal villi and intestinal crypt structures [34]. Using a similar system, Kim et al. co-cultured human intestinal epithelial cells, Caco-2 cells, and intestinal bacteria (*Lactobacillus rhamnosus*) on an intestine-on-a-chip that simulated the peristaltic movement and fluid flow of the intestinal lumen, with the co-culture of *L. rhamnosus* and epithelial cells on the chip extending for more than 2 weeks [35]. Compared with cells grown using traditional culture methods, those grown on chips can more closely simulate the barrier function of the intestinal villi, cytochrome P450 activity, and apical mucus secretion.

The intestine-on-a-chip also provides a platform for studying the pathogenic mechanisms of different microorganisms. Using an intestine-on-a-chip, Villenave et al. introduced coxsackievirus B1 (CVB1) to infect villi of the small intestine and reported active replication and apical release of virus particles in epithelial cells, along with cytopathic effects and the production of inflammatory cytokines (IP-10 and IL-8) [36]. The apical release of viral particles on a chip has been similarly reported for Sendai virus [37], human parainfluenza virus type 3 (HPIV3) [38], respiratory syncytial virus [39], and the viruses causing measles [40], mumps [41], and hepatitis A [42]. With respect to bacterial infection, Grassart et al. successfully replicated *Shigella flexneri* infection on a chip [43] and showed that bacteria successfully accumulated in the intestinal crypts formed on the chip and invaded the intestinal epithelial villi, thereby resulting in a loss of structural integrity. Notably, the cyclic strain and shear stress in the intestinal microenvironment was found to enhance the invasive efficacy and transmission of *S. flexneri* [43]. Furthermore, Tovaglieri et al. constructed a dual-channel colon chip lined with primary human colon epithelial cells [44], to which they introduced metabolites of mouse or human intestinal microbiota harboring the metabolites enterohemorrhagic *Escherichia coli* (EHEC). The authors showed that human microbial metabolites enhanced EHEC-induced intestinal epithelial injury, thereby providing evidence that intestine-on-a-chip technology has broader application for translational research studies.

## 4. Advantages and Disadvantages of Intestine-on-a-Chip

The intestine-on-a-chip system simulates the intestinal environment in a manner that preserves the spatial structure of intestinal epithelial tissue, the interaction between epithelial cells, the interactions between epithelial cells and microorganisms, and peristalsis of the intestinal tract, thereby enabling the accurate replication of a range of different physiological conditions [45]. The physiological microenvironment of a given intestine-on-a-chip is collectively established by the integration of different characteristics, including the 3D intestinal villous structure, multi-lineage cell differentiation, epithelial barrier function, brush border enzyme activity, and mucus production. Moreover, compared with Caco-2 cells cultured using traditional methods, cells cultured on an intestine-on-a-chip are more similar to human duodenal epithelial cells in terms of morphology, structure, multicellular composition, and gene expression patterns [31]. Moreover, these chips can be integrated with in situ visualization instrumentation, such as two-photon or rotating disk microscopes, to visually monitor the co-culture conditions of intestinal epithelium and microorganisms. This coupling is achieved by continuously collecting aliquots of liquid flowing through the chip channels and quantitatively detecting metabolites that reflect the status of a diverse range of processes and conditions, including digestion, secretion, intestinal barrier function, and microbial infection [46].

The intestine-on-a-chip platform provides a controlled micro-platform that facilitates the study of the complex associations between the intestine and microorganisms based on relevant pathophysiological indices. These include the independent or combined effects of microorganisms, lipopolysaccharides (LPSs), immune cells, inflammatory cytokines, and mechanical forces on the maintenance of intestinal homeostasis, which can contribute to elucidating the mechanisms underlying intestinal inflammation, villous damage, and the damaged barrier function of epithelial cells. In addition, the physiological or pathological activities relating to specific areas of the intestinal tract can be simulated by introducing agents such as immune cells or microorganisms to the intestine-on-a-chip to enable a prospective analysis of different intestinal diseases [47].

Despite these multiple advantages, however, these chips have a high technical threshold and a slow, complex process that is not well-suited to rapid analyses. Although some progress has been made in combining bacterial cultures with gut-specific microarrays, the small size of these chips, slow flow rate, and complex technology required to meet the necessary oxygen demand tends to limit the broader application of these chips [48]. In addition, the continuous co-culture of microbiota and cells has yet to be realized using existing intestine-on-a-chip platforms, nor can they integrate the regulatory effects of the endocrine and nervous systems, which renders these systems unsuitable for studying chronic intestinal diseases.

## 5. Application of Microfluidic Drug Delivery Systems

Drug delivery systems, among which are nanoparticles [49], gel complexes [50], and enteric coating, are routinely applied in research on intestinal microorganisms [51]. As an advanced technology, microfluidics can be used to synthesize functional microcapsules with tailored size, morphology, and structure to deliver active payloads [52,53,54,55]. The drug delivery systems based on this technology can facilitate drug pre-programming and controlled drug delivery, and can be applied to supplement the human body with prebiotics, probiotics, and other active substances [56]. Among microfluidic drug delivery systems, droplet microfluidics uses a fluid to provide a driving force that manipulates liquid droplets, whereas microfluidic electrospray technology utilizes static electricity to manipulate these droplets [57].

Droplet microfluidic technology can be used for delivering active bacteriophages into the intestine, wherein these viruses can influence the intestinal flora by disrupting intestinal bacteria, regulating the immune response, and mediating anti-inflammatory responses. Bacteriophages are an important component of the intestinal microbiota [58], and it has been reported that high-dose phage samples show better antibacterial effects [59,60,61,62,63,64]. Phage delivered via the oral route are typically exposed to gastric acid, bile, and digestive enzymes, which collectively contribute to reducing the phage titer [65], thus reducing the efficacy of phage therapy [66,67]. As shown in Figure 2A, Vinner and Malik used droplet microfluidic chips to microencapsulate bacteriophages, thereby facilitating transport in the active condition [68]. To produce a continuous oil phase, Miglyol 840, a propylene glycol diester of caprylic/capric acid, and castor oil were mixed in a 50:50 ratio, along with 5% (*w*/*v*) of the oil-soluble surfactant polyglycerol polyricinoleate (PGPR). The dispersed phase consisted of a mixture of the methyl methacrylate co-methacrylic acid copolymer Eudragit^®^ S100, alginate, and bacteriophages. Using a pressure pump, the two immiscible fluids were brought together at the perpendicular intersection of a T-junction channel, and under the influence of pressure and shear forces, the flowing phase fragmented the dispersed phase, forming droplets, as shown in Figure 2B [69]. Figure 2C shows a real-time micrograph of droplet generation. The resulting emulsion was collected in an acidified oil phase containing 0.05 M *p*-toluenesulfonic acid in Miglyol and 5% (*w*/*v*) PGPR, after which the emulsion was left to crosslink for a minimum of 2 h.

Having settled and been acidified by the application of acidified oil, the oil layer and the acidified oil are removed from the liquid drops, and calcium chloride (CaCl_2_) is added to induce ionic crosslinking and generate microcapsules, as shown in Figure 2D Exposure of these microcapsules to simulated gastric fluid (pH = 1) for 2 h revealed that there was no reduction in phage titer, and the subsequent release of these bacteriophages in simulated intestinal fluid (pH = 7) verified that this pH-induced microcapsule release is suitable for targeted delivery of active microorganisms to the gastrointestinal tract. Moreover, the rate of bacteriophage release (or that of other payloads) can be adjusted by controlling capsule size. Bacteriophages can be wrapped in microcapsules of differing size (50–100 μm), with more rapid release being achieved using smaller capsules. In contrast, whereas larger capsules are also characterized by rapid release during the early stage of delivery, release is slower and more sustained during the latter stages.

When combined with probiotics, microfluidic electrospray technology can be applied for the treatment of metabolic syndrome (MetS). Probiotics are active microorganisms that are beneficial to host health, playing important roles in maintaining gut health by correcting intestinal flora imbalance and protecting against pathogen invasion, which is considered an important strategy for MetS treatment [72,73]. In this context, Zhao et al. used microfluidic electrospray chips to produce binuclear probiotic microcapsules, in which alginate was used as an external phase and the microspheres containing two species of probiotic bacteria (*Lactobacillus* and *Bacillus subtilis*) were electrosprayed as an internal phase (Figure 2E) [71]. Ionic cross-linking of the sodium alginate encapsulated the probiotic microspheres within a core-shell structure, which was further covered in a dietary fiber shell that protects the contents from gastric juice attack. The alginate shell was pH-responsive, and, consequently, whereas the contents of these microcapsules are protected from the influence of gastric acid in the stomach, they are efficiently released on entering the intestinal lumen (Figure 3). A mouse model of MetS was generated based on feeding mice a high-fat diet, and these animals exhibited the characteristics of abnormal liver fat metabolism, loss of intestinal barrier proteins, and deformed intestinal villous morphology [74]. Mice to which the probiotic microcapsules had been administered, via gavage, were found to have reduced liver fat deposits and enhanced levels of intestinal barrier proteins. These findings thus indicate that such microcapsules can be used to protect the intestinal barrier, reduce liver fat deposition, and improve MetS. Moreover, such dual-core microcapsules enabled the use of different combinations of probiotic organisms, thereby enabling the exploitation of synergistically beneficial effects.

Microfluidic electrospray techniques can also be used to generate microcapsules containing detoxified LPS, an outer membrane component of gram-negative bacteria and a causative factor of multiple diseases and disorders, including sepsis, septic shock, and MetS [75,76,77,78]. Within the human body, LPS is mainly derived from intestinal microorganisms [79]. The intestinal barrier plays an important role in combating the harmful effects of different microbial toxins, including LPS [80,81,82], which is inactivated by the action of alkaline phosphatase, thereby playing a vital role in the maintenance of intestinal homeostasis and disease prevention [83,84,85]. The intestinal degradation of microcapsules leads to the release of their contents into the intestine, which thereby contributes to strengthening the bionic barrier, as confirmed by internal computerized tomography imaging and in vivo imaging system analysis [86]. Furthermore, microcapsule contents can be released along the intestinal tract and form a continuous barrier, which in turn blocks LPS permeation and its detrimental effects on intestinal epithelial cells. This effectively suppresses the influence of harmful intestinal microorganisms on the intestinal mucosa, thereby contributing to the maintenance of enteric health [87]. Table 2 summarizes the applications of different microfluidic platforms in gut microbiota research.

## 6. Advantages and Disadvantages of Microfluidic Drug Delivery Systems

The emergence of microfluidic technology has radically transformed the manufacture and development of microcapsules [89]. Among the available techniques, droplet microfluidic technology is most commonly applied for synthesizing drug carriers and uniform microcapsules [90]. Through a combination of structural design, physical and chemical processes, and functional components, these microcapsules can conveniently carry living intestinal microorganisms. Furthermore, in microfluidic electrospray systems, precise control enables the microbial loading of microcapsules and subsequent release, thereby providing ideal delivery systems [91,92]. Moreover, microfluidic electrospray technology can be used to establish multilayer packaging for protecting the bioactive contents [93,94]. Microfluidic electrospray has thus become one of the commonly used techniques for the manufacture of functional microspheres or microcapsules [95], which can contribute to solving problems such as low biological activity, poor stability, and the uncontrollable release of bioactive contents, thereby enhancing their bioavailability and efficacy.

However, despite the aforementioned advantages, microfluidic drug delivery systems require several components, such as microfluidic chips, high-precision pressure pumps, rotary valves, microvalves, flow meters and manometers, microscopes, microzone spectrometers, and impedance meters, that increase their technical thresholds. In addition, the efficiency of drug release is affected by multiple factors, including the preset parameters, equipment used, wrapping materials, and individual differences, which tend to hinder advances in the research on intestinal microorganisms. Consequently, to enhance the efficacy and cost-effectiveness of microfluidic drug delivery systems, it will be necessary to further study the parameters and materials that affect drug release, develop more efficient microfluidic systems, and examine the potential utility of microfluidic technology in intestinal microbe research. By facilitating screening and optimizing the delivery of intestinal microorganisms and packaging of multiple microorganisms, microfluidic electrospray can provide a basis for research on the effects of a diverse range of microorganisms in the human intestine, as summarized in Table 3.

## 7. Discussion

Microfluidic technology can be applied to accurately control the characteristics of different fluids, and as such, it has emerged as a prominent interdisciplinary research field [96]. The development of microfluidic technology will facilitate the construction of complex intestinal disease models, the delivery of active microorganisms, and drug development [97]. In the future, it is anticipated that intestine-on-a-chip systems will replace costly and inefficient sterile tissue culture or traditional animal models for the study of intestinal–microbiota interactions. As intestine-on-a-chip technology advances, the cost of chip fabrication can be further reduced, chip manufacturing technology can be improved, and more advanced chip designs can be implemented to achieve a more effective simulation of the intestinal environment, thereby facilitating continuous co-culture of the gut microbiota and cells. Moreover, these advances will potentially contribute to the development of systems combining intestine-on-a-chip technology with other technologies, such as metabolomics, proteomics, and gene expression profiling, which is predicted to enable us to gain a better understanding of the intestinal microenvironment and the complex interactions among gut microbiota. In addition, studies on the integration of the gut with other systems, such as the nervous and endocrine systems, based on the intestine-on-a-chip, and the development of new technologies for the continuous monitoring of intestinal epithelial and microbial co-culture conditions, will contribute to a more comprehensive understanding of the pathogenesis of intestinal diseases, the development of personalized prevention and treatment strategies for different intestinal diseases, and greater insights into the mechanisms underlying chronic intestinal diseases. Intestine-on-a-chip-based analyses will thus become an indispensable approach for studying the complex interactions between the gut microbiota and the human body.

In tandem with the development of material science, microfluidic drug delivery systems will give rise to the production of multifunctional microcapsules that can encapsulate a larger number of microorganisms and exploit a diverse range of trigger-release strategies, including temperature, pH, and enzymic, which will contribute to more effectively solving the current problems of low bioactivity, poor stability, and uncontrollable drug release. Moreover, these systems can provide a basis for establishing polymicrobial studies via screening and optimization of the delivery and transport of intestinal microorganisms, in addition to systems that facilitate the delivery of multiple microbes. Furthermore, microfluidic drug delivery systems will also enhance the efficacy of oral drug delivery, shorten treatment times, and improve the quality of treatment by enhancing the precision and efficiency of structural designs, physical and chemical processes, and functional components. In addition, new materials or technologies can be applied to increase the capability and quality of microfluidic drug delivery systems, such as biodegradable materials, multilayer encapsulation, and reactive oxygen species inhibitors, and to continue optimization of the technology and enable cost reductions. This will ultimately enhance the performance and efficiency of microfluidic drug delivery systems and lower the technological threshold for examining the potential of microfluidic technology in the study of intestinal microorganisms.

The application of microfluidic technology in the study of intestinal microorganisms will also contribute to accelerating research on the mechanisms underlying intestinal diseases and the development of related drugs. Finally, complex intestinal physiological and pathological phenomena, such as those associated with intestinal barrier functions [98], inflammatory bowel disease [99], and intestinal infection, will be identified, thereby presenting opportunities for the better prevention and control of diseases.

## Figures and Tables

**Figure 1 microorganisms-11-01134-f001:**
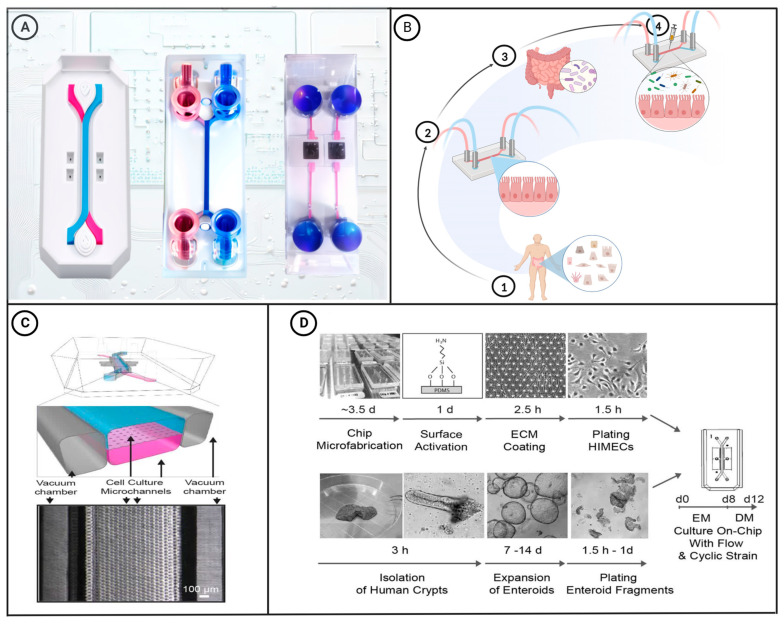
(**A**) Schematic diagram of commercially available microfluidic chips. The diagrams on the left and middle show two-channel organ-on-a-chip devices. These chips can be used to create colon-intestine chips, duodenum-intestine chips, liver chips, and other types of organ chips [29]. The two-channel microfluidic chip on the right connects the culture well in its center to the microfluidic channel through a porous membrane. This type of chip is commonly used for research on gas–liquid interfaces and the endothelial–epithelial barrier. (**B**) Construction of a microfluidic intestinal chip with the capacity for microbial co-culture: ① Intestinal epithelial cells are extracted from the human body via biopsy. ② The obtained epithelial cells are then cultured in a microfluidic intestinal chip. ③ Gut microbiota required for the experiment are extracted. ④ The microbiota are introduced into the microfluidic intestinal chip for co-culture with the epithelial cells. [30]. (**C**) A schematic cross-sectional view (top) and a phase contrast micrograph of the intestine chip viewed from above (bottom) showing the upper (epithelial; blue) and lower (microvascular; pink) cell culture microchannels separated by a porous, ECM-coated, PDMS membrane sandwiched in between. The membrane is elastic and can be extended and retracted by the application of a cyclic vacuum to the hollow side chambers. This vacuum actuation results in outward deflection of the vertical side walls and lateral extension of the attached horizontal porous elastic membrane, leading to mechanical deformation of the adherent tissue layers cultured in the central channels [31]. (**D**) This figure shows a step-by-step schematic representation of the procedure for establishing microfluidic co-cultures of primary human intestinal epithelium and intestinal microvascular endothelium in the Intestine Chip [31]. Figure 1C,D have been edited and modified with permission from [31]. Copyright 2018 Authors, licensed under a Creative Commons Attribution 4.0 International License.

**Figure 2 microorganisms-11-01134-f002:**
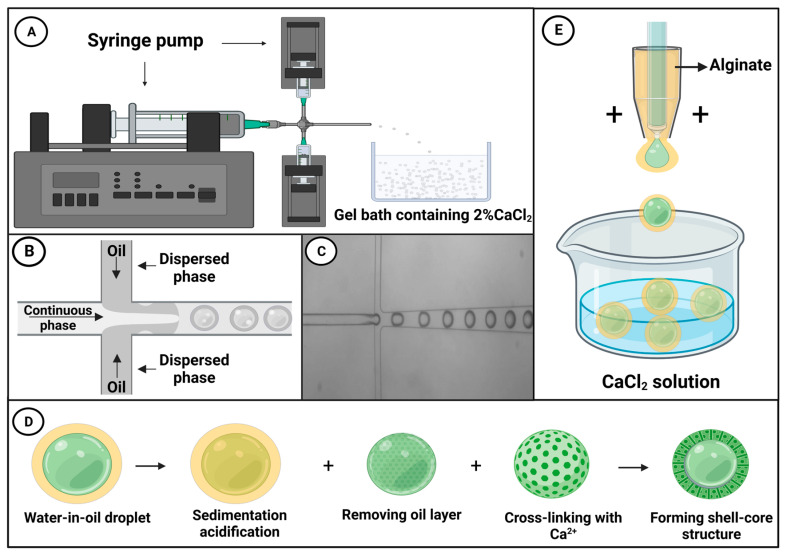
(**A**) A simple diagram showing a droplet microfluidic system, in which a droplet microfluidic chip is used to generate droplets that are subsequently cured in a 2% CaCl_2_ gel to yield microcapsules [68]. (**B**) A schematic diagram of droplet microfluidic technology [69]. (**C**) An optical micrograph of the microparticle encapsulation process. (**D**) Following water-in-oil particle settlement and acidification using an acidified oil, the oil layer and the acidified oil are removed from the droplets, after which CaCl_2_ is added to crosslink and solidify the alginate ions, thereby generating microspheres with a core–shell structure [70]. (**E**) A schematic diagram of a device used to produce microcapsules via microfluidic electrospray [71].

**Figure 3 microorganisms-11-01134-f003:**
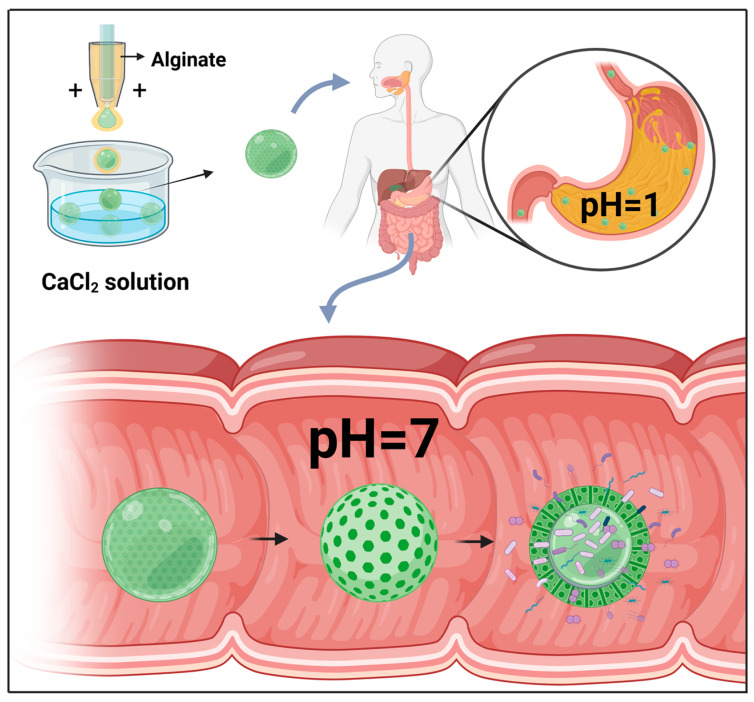
Oral delivery of microcapsules produced using a liquid droplet microfluidic chip. Given the specific pH-induced release characteristic of these microcapsules, they are protected from attack by gastric fluid, whereas the contents of active microorganisms are effectively released within the intestinal fluid.

**Table 1 microorganisms-11-01134-t001:** A summary of the history of microfluidic technology development.

Year	Material	Platform	Method	Result
1990	Glass	Total chemical analysis system	Standard photolithography and etching processes	Microfluidic technology was proposed for the first time, and capillary electrophoresis was performed on a flat microchip the following year [12]
1994	Glass	Glass microchip column	Change in the injection scheme and column geometry of the microchip	The performance and practicability of capillary electrophoresis of the microfluidic chip was improved [13]
1995	Silicon dioxide	Capillary array electrophoresis chips	DNA sequencing on a microchip	High-speed DNA sequencing on microchips [14]
1996	Silicon and glass	Integrated DNA analysis system	Microfabricated silicon PCR reactors and glass capillary electrophoresis chips coupled to form an integrated DNA analysis system	PCR and capillary electrophoresis were simultaneously integrated on a microfluidic chip [15]
2002	Silicon	Microfluidic multiplexors	Microfluidic chips integrating thousands of microvalves and microreactors	The leap from simple electrophoresis to a large multifunctional integrated laboratory was realized [17]
2004	Dimethylsiloxane	Microfluidic channels	Reliance on chaotic advection to rapidly mix different reagents dispersed in droplets	The emergence of droplet-based microfluidic technology [18]
2010	Polydimethylsiloxane + polyethylene glycol diacrylate	Droplet microfluidic device based on polydimethylsiloxane	Polyethylene glycol diacrylate hydrogel beads encapsulating *Escherichia coli* were prepared	Development of a new polymerization technique using a microfluidic device to fabricate monodisperse hydrogel microbeads [19]
2011	Hydrogel	Microfluidic chip	To co-culture symbiotic microbial communities, highly parallel droplets were utilized in the study	Uniform-sized hydrogel microbeads were successfully fabricated. Intestinal microorganisms were encapsulated in hydrogel microbeads with high efficiency [20]
2015	Hydrogel	Microfluidic chip	Microscale culture chambers were created in microfluidic chips using hydrogel structures	A range of bacterial strains were successfully encapsulated and co-cultured in a microculture chamber fabricated on a microfluidic chip using hydrogels [21]
2017	polydimethylsiloxane (PDMS) and Alginate	Microfluidic electrospray	Complex particle engineering was achieved using tri-needle coaxial electrospraying	A microfluidic chip with a PDMS microwell array was successfully fabricated. Bacteria were encapsulated in alginate droplets within the PDMS microwell array. The co-culturing of different bacterial strains was achieved within the alginate droplets [22].
2021	Polydimethylsiloxane and okara	Droplet microfluidic device based on polydimethylsiloxane	Probiotics were encapsulated in an emulsion consisting of okara oil and polyacrylic acid	Using polyacrylic acid to package probiotics, the activity of probiotics could be preserved when in contact with the gastrointestinal tract [23]

**Table 2 microorganisms-11-01134-t002:** A summary of the applications of different microfluidic platforms in gut microbiota research.

Research Platform		Cell Types	Microbial Types	Experimental Results	Significance of Experiment
Microfluidic gut-on-a-chip		Intestinal epithelial cells, endothelial cells, Caco-2 cells, absorptive cells, mucus-secreting cells, enteroendocrine cells, Paneth cells	Lactobacillus rhamnosus, Coxsackievirus B1, Shigella flexneri enterohemorrhagic Escherichia coli	1. Simulation of intestinal villous barrier function [88]; 2. Simulation of biomechanical properties of intestinal peristalsis and fluid flow, formation of small intestinal villous structure, and co-culture of intestinal epithelial cells with Lactobacillus rhamnosus for several days to weeks [35]; 3. Successful replication of viral infection [36]; 4. Successful replication of Shigella flexneri infection [44].	Used to study the interaction between intestinal cells and microbes in the gut microbiota, simulate the gut environment, investigate the intestinal absorption function and barrier function, and evaluate the impact of gut microbiota on human health.
Microfluidic drug delivery system	Droplet microfluidic platform	None	Bacteriophages	Droplet microfluidic technology can be used to deliver active bacteriophages to the gut [68].	Used to study microbial therapy, microbial delivery, and microbial release technology, and examine novel microbial therapy strategies.
	Microfluidic electrospray platform	None	Probiotics	1. Microfluidic electrospray technology combined with probiotics can prepare microcapsules for the treatment of MetS [71]. 2. Microfluidic electrospray technology can also be used to prepare microcapsules containing detoxified lipopolysaccharide, promote gut health by enhancing biomimetic barriers, and prevent harmful microbes from affecting the gut mucosa [86].	Used to study microbial therapy, microbial delivery, and microbial release technology, and explore novel microbial therapy strategies. It can also be used to study the construction of biomimetic barriers and gut barriers, and assess new methods for protecting gut health.

**Table 3 microorganisms-11-01134-t003:** A comparison of the application of two microfluidic platforms with traditional microbiological platforms (culture of bacteria and 16S rRNA sequencing) in microbiological studies.

Platforms		Required Equipment	Advantages	Limitations
Microfluidic intestinal chip		Microscope, microfluidic chip fabrication equipment, microfluidic chip, temperature-controlled cell culture chamber, high-precision temperature controller, humidity control module, micro-volume precision syringe pump, gas mixer, multi-channel reagent automatic switching valve, fully automated cell perfusion system	1. Microfluidic chips can better simulate the growth environment of microorganisms in vivo, with higher accuracy and controllability. 2. Microfluidic chips can establish multiple channels within the chip, enabling different types of microorganisms to be co-cultured on the same chip, thereby better assessing the interactions between microorganisms. 3. Microfluidic chips can achieve co-cultivation of microorganisms with host cells, thereby better simulating the interaction between microorganisms and hosts.	Requires specialized equipment and expertise.
Microfluidic drug delivery system	Microfluidic electrospray technology	Microfluidic chip fabrication equipment, microfluidic electrospray chip, high-voltage power supply, syringe pump, spray needle, solvent reservoir, collection substrate	1. Precise control of droplet size and production rate. 2. High drug loading capacity. 3. Reduced solvent consumption.	Limited by the quality and quantity of the drug solution.
	Droplet microfluidics	Computer, microfluidic chip fabrication equipment, droplet chip, chip holder, liquid storage tank, syringe pump or pressure controller, flow sensor, droplet	1. Precise control of droplet size and production rate. 2. High drug loading capacity.	1. Requires specialized equipment and expertise. 2. Limited by the quality and quantity of the drug solution.
Culture of bacteria		Culture medium, incubator, Petri dishes, bacterial strains	Well-established and widely used.	1. Time-consuming. 2. Limited diversity of microbiota.
16S rRNA sequencing		DNA extraction equipment, PCR device, sequencing device	Can identify and classify microorganisms based on their DNA sequences.	Limited by quality and quantity of DNA samples. Does not provide information on the functional properties of microbiota.

## Data Availability

This is a review article and does not contain any research data.

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
