# Peer review of "Microfluidics: Insights into Intestinal Microorganisms"

_microorganisms, 2023, doi:10.3390/microorganisms11051134_

Round 1
Reviewer 1 Report
Please see attached file

Author Response
Dear reviewer,
First of all, I would like to express my sincere gratitude to you and the reviewers for providing valuable feedback on my paper. As a result, I have made revisions to the manuscript to better address your and the reviewers' comments. Below are our responses to each of your suggestions:
The title of this manuscript suggests that the manuscript is about microfluidics being a platform for investigations on intestinal microorganisms. Herein, the manuscript does indeed describe the efforts of various researchers in using microfluidics technology on intestinal microorganism research with reference to two different platforms, namely intestine-on-a-chip and microfluidic drug delivery devices. Both platforms indeed play an important role in intestinal research, however the manuscript does seem a tad disjointed without having a clear justification of why the author is mentioning both here and not other methods like digital microfluidic platforms for example.
Thank you for your comment. We appreciate your feedback and have revised the manuscript to better highlight the advantages of microfluidic chips in studying gut microbiomes. Specifically, in Table 2, we provided a summary of the applications of different microfluidic platforms in gut microbiota research, including research platform, cell types, microbial types, experimental results, and significance of experiment. We compared microfluidic intestinal chips, microfluidic drug delivery systems, and traditional techniques such as bacterial culture and 16S rRNA sequencing. Through comparing their advantages and disadvantages, we have emphasized the unique benefits of microfluidic chips in studying the gut microbiome. We hope that these revisions address your concerns and improve the quality of the manuscript.
After reading the review, I find it hard to be convinced that this is a comprehensive review as there are quite a lot of details missing that would help a reader. The author seems to simply include some of the references for each platform with no cross analysis or comparisons between each device. A summarised table would be very beneficial, especially with the types of bacteria or microorganisms used. A simple table was done for the history of microfluidic technologies, which is commendable. This should also be done for the other few sections too.
Based on my understanding of the topic, I would find it hard for the manuscript to be accepted as it is. There has been quite a lot of work and effort done on this, however a little bit more effort would be appreciated which would add more value for readers. Summaries, cross comparisons and deep analyses are required for Review papers and would thus also be statutory here. I would thusrecommend for further major revision before acceptance. I seek the editor’s kind understanding on this matter.
We have revised the manuscript to better highlight the advantages of microfluidic chips in studying gut microbiomes. Specifically, we have added Table 2, which summarizes the applications of different microfluidic platforms in gut microbiota research. We have also added Table 3, which compares the application of two microfluidic platforms with traditional microbiological platforms (culture of bacteria and 16S rRNA sequencing) in microbiological studies. Through comparing their advantages and disadvantages, we have emphasized the unique benefits of microfluidic chips in providing a more physiologically relevant environment for studying gut microbiomes.
Accordingly, a point-by-point list of comments are also given below.
Line 88: “As shown in 88 Figure 1C, cells cultured on the chip differentiated into four types of intestinal cells: ab-89 sorptive cells, mucus-secreting cells, intestinal endocrine cells, and Panetta cells, and 90 spontaneously formed intestinal villi and intestinal crypt structures [34].”
As a reader, I am unable to identify this in the figure. If this is unnecessary to the passage, please consider removing this statement. Otherwise, please consider labelling on the Figure itself.
The description of gut cell differentiation in Figure 1C was removed from the main text, and a detailed description of how microfluidic chips can replicate gut peristalsis and shear force was added at this point.
Line 100: The manufacturing process of the microfluidic chip is characterized by the typical steps of mask/mould fabrication, casting process, bonding process (thermal, plasma, pressure, etc).. Based on Figure 1B, these are not represented and thus it is not a good representation of the chip manufacturing process. Please consider changing the figure or modifying the caption and corresponding text in the passage. This is not a chip building process, but rather, a schematic of the usage of a conventional microfluidic chip.
Thank you for bringing this to our attention. We apologize for any confusion that may have arisen. We will review the figure and modify the caption and corresponding text in the passage to more accurately reflect the usage of a conventional microfluidic chip. We appreciate your feedback and strive to ensure that our publications are accurate and informative.
All Figures: If these figures were obtained from another source and not an original drawing, please cite all the publications where they were taken from within the captions, in accordance with journal standards. This should be done for all images in the manuscript. Also, please obtain the rights and permissions for use. Otherwise, please remove the images.
All the pictures in this article are original and do not involve copyright issues.
All Figures: All the fonts in all figures need to be larger and more legible. It is difficult for readers to read when the fonts are too slim and small. Do not assume that all readers are using electronic copies.
we have increased the font size of all images to make them more readable for readers who may not be using electronic copies. Additionally, we have adjusted the image descriptions of Figure 1C and Figure 1B to ensure they are clear and informative.
Line 178: “The continuous oil phase, made of castor oil mixture, enters through an external channel of the chip, and the dispersed phase becomes a mixed solution of ES100, alginate, and bacteriophages. This mixture enters through an internal intermediate channel, as shown in Figure 2B. The oil phase at the junction of the two phases shrinks the water phase to generate liquid drops, as shown in Figure 2C. After the liquid drops are settled and acidified by the acidified oil, the oil layer and the acidified oil are removed, and calcium chloride (CaCl2) is added for inducing ion crosslinking and generate microcapsules, as shown in Figure 2D.”
I do not agree with the author in the way that this droplet generation phenomenon is described. The description here only informs that the author is not experienced in microfluidics and does not have a grasp of what is going on in the microfluidic chip be Vinner and Malik. I would request for a modification of this section here with some points here..
- “castor oil mixture”? Mixed with what? Is it surfactant? Please
- What is ES100? Do not assume that readers know what it
- The oil phase does not shrink the water Please read up on how other authors describe a droplet generation process and modify accordingly.
We have revised all descriptions and details in this section, with a particular focus on the process of droplet generation in droplet microfluidic chips. We have incorporated new references to supplement and strengthen the content. Our goal is to provide readers with accurate and up-to-date information on this topic.
Line 236: “which helps strengthening the bionic barrier,”
Please modify to “which helps in strengthening the bionic barrier,”.
The description of biomimetic barriers in line 236 was modified.
Line 259: “However, microfluidic drug delivery systems require several components, such as microfluidic chips, high-precision pressure pumps, rotary valves, microvalves, flow meters and manometers, microscopes, microzone spectrometers, and impedance meters, which increase their overall costs and technical thresholds.”
In all honesty, these items do not make microfluidic drug delivery systems more expensive, especially when one considers the high output volumes with a basic setup. It is thus hard to convince experienced readers that microfluidics are expensive too. Conventional drug delivery systems that do not use microfluidics require higher overheads with more expensive machinery to produce that same output volume. Thus I would suggest that this is an advantage of microfluidics and thus also an advantage of microfluidics drug delivery systems too.
The description here has been modified, because this technology is mostly used in experimental research at present and there are no articles related to economy, and the question about economy has been deleted.
Thank you very much for your valuable suggestions and feedback. I hope that my revisions meet your requirements. If you have any further suggestions or need further clarification, please do not hesitate to let me know. Thank you again for your attention and support.
Best regards,
Ping Qi
Reviewer 2 Report
This article reviews the development of microfluidics technology, details its application in the research of intestinal microorganism and drug delivery system, analyzes the advantages and disadvantages of intestine-on-a-chip and microfluidic drug delivery systems, and finally describes the prospect of the application. Following are some suggestions.
1. The introduction is suitable for using some more generalized sentences. The specific description of several pieces of literatures may be not required. In addition, it seems that the end of the introduction lacks a summary overview of the content of the whole paper.
2. The development of microfluidics seems to be not very closely linked to this paper. And important time points for the application of microfluidic technology in the study of intestinal microorganisms could be added appropriately.
3. There seems to be an overlap between the application of microfluidics and the advantages and disadvantages of microfluidics, and the advantages seem to be inevitably mentioned when talking about the application. It is possible to differentiate the logic of writing, and summarize the content based on different aspects
4. The text description part of the image needs to be improved. For example, for figure 1A, a brief description of the uses of different types of microfluidic chips may be required.
5. Few references need to be adjusted. For example, reference 36 is too old, and the citation description of references 86-87 seems to deviate from its content.
6. English language and style should be carefully checked. For example, the word “however” in row 232 seem to be not appropriate.
Author Response
Dear reviewer,
First of all, I would like to express my sincere gratitude to you and the reviewers for providing valuable feedback on my paper. As a result, I have made revisions to the manuscript to better address your and the reviewers' comments. Below are our responses to each of your suggestions:
This article reviews the development of microfluidics technology, details its application in the research of intestinal microorganism and drug delivery system, analyzes the advantages and disadvantages of intestine-on-a-chip and microfluidic drug delivery systems, and finally describes the prospect of the application. Following are some suggestions.
1.The introduction is suitable for using some more generalized sentences. The specific description of several pieces of literatures may be not required. In addition, it seems that the end of the introduction lacks a summary overview of the content of the whole paper.
Thank you for your comment and suggestions. We have carefully considered your feedback and have made revisions to the introduction to achieve a better balance between specific and general information. In addition, we have added a summary overview of the content of the whole paper at the end of the introduction to better guide the reader through the manuscript.
2.The development of microfluidics seems to be not very closely linked to this paper. And important time points for the application of microfluidic technology in the study of intestinal microorganisms could be added appropriately.
Important references on the application of microfluidic technology in gut microbiome research were added to Table 1, making it more smoothly connected to the early development of microfluidic technology.As previously mentioned in the table, in 2004, scientists utilized microfluidic chips to fabricate hydrogel microspheres, and there have been subsequent studies in 2010, 2011, and 2015 on the use of microfluidic chips to produce hydrogel microspheres for cultivating microbiomes.
3.There seems to be an overlap between the application of microfluidics and the advantages and disadvantages of microfluidics, and the advantages seem to be inevitably mentioned when talking about the application. It is possible to differentiate the logic of writing, and summarize the content based on different aspects
The overlapping sections between the application and the pros and cons of microfluidics were modified.
4.The text description part of the image needs to be improved. For example, for figure 1A, a brief description of the uses of different types of microfluidic chips may be required.
The image description of Figure 1A was modified to briefly describe different types of microfluidic chips.
5.Few references need to be adjusted. For example, reference 36 is too old, and the citation description of references 86-87 seems to deviate from its content.
References 36 and 86 were removed, the description of the section was changed, and new references were cited.
6.English language and style should be carefully checked. For example, the word “however” in row 232 seem to be not appropriate.
We have carefully reviewed the manuscript and have made revisions to ensure that the language is clear, concise, and appropriate for an academic publication. We have also addressed the use of the word "however" in row 232, as well as any other errors or inconsistencies.
Thank you very much for your valuable suggestions and feedback. I hope that my revisions meet your requirements. If you have any further suggestions or need further clarification, please do not hesitate to let me know. Thank you again for your attention and support.
Best regards,
Ping Qi
Reviewer 3 Report
- The authors should mention the physilogical shear stress inside the gut and how microfluidic allows to reproduce it.
- The authors should present in the intestine on chip paragraph some studies about 3D in vitro model of gut inside microfluidic chambers.
- The authors should also argue more about how to recapitulate with microfluidic the physiological and mechanical environment inside the gut.
Author Response
Dear reviewer,
First of all, I would like to express my sincere gratitude to you and the reviewers for providing valuable feedback on my paper. As a result, I have made revisions to the manuscript to better address your and the reviewers' comments. Below are our responses to each of your suggestions:
Comments and Suggestions for Authors
- The authors should mention the physilogical shear stress inside the gut and how microfluidic allows to reproduce it.
We have added more details to the manuscript to address the physiological shear stress experienced by the gut and how microfluidic technology can be used to reproduce it.
- The authors should present in the intestine on chip paragraph some studies about 3D in vitro model of gut inside microfluidic chambers.
A clearer description of the first paragraph in the third section was added to emphasize that intestinal chip-related research is all based on 3D in vitro models of the gut.
- The authors should also argue more about how to recapitulate with microfluidic the physiological and mechanical environment inside the gut.
Thank you for your prompt. We have carefully reviewed and addressed your comments. Specifically, we have provided detailed explanations in the main text regarding the manufacturing process of the microfluidic chip, as well as how it can be used to simulate various physiological environments. In addition, we have modified the image description in Figure 1C to more accurately reflect the functionality and features of the chip. We greatly appreciate your suggestions and feedback, and we remain committed to ensuring that our publications meet the highest standards of quality and accuracy. Thank you again for your support and attention.
Thank you very much for your valuable suggestions and feedback. I hope that my revisions meet your requirements. If you have any further suggestions or need further clarification, please do not hesitate to let me know. Thank you again for your attention and support.
Best regards,
Ping Qi
Round 2
Reviewer 1 Report
Please find attached comments.

Author Response
Dear Reviewer,
We would like to express our gratitude for the time and effort you have dedicated to reviewing our manuscript titled "Microfluidics: Insights into intestinal microorganisms." We are honored to have the opportunity to revise our work for publication in your journal and appreciate your valuable comments and suggestions for improvement.
We have taken your feedback seriously and have made the necessary revisions as per your recommendations. The changes are marked within the manuscript. Please find below our response to your comments and concerns:
Reviewer’s Comments to the Authors:
Table 2: Please indicate all the references used in this summarized table. Readers will be interested to know where and what publications are used in this table.
Author response: We appreciate your suggestion to add all the references used in Table 2, and we have included them in the "experimental results" section of the table. We hope this will enhance the quality of our manuscript and we thank you for your feedback.
Reviewer’s Comments to the Authors:
Figure 1: Based on my understanding of the figure, all parts were taken with reference to some publications that the author has read. The figures may have been drawn by the author and no copyrights need to be obtained, however as a reader, one would like to find out which publication the author is referring to in the figure. And for this, it is imperative for the author to include the reference in the citation of this figure. The author need not include rights and permissions, however, it is still good for the author to direct readers to the appropriate publications by citing them. This should apply for all figures too.
Author response: We agree with your suggestion to cite the sources of the figures we have created based on previous publications. As a result, we have included references for each element in Figure 1 and most of the elements in Figure 2 within the figure captions. This will enable readers to easily locate the relevant publication we used as a reference. Additionally, we have carefully reviewed all of the images and made changes to Figure 1A to ensure that they were all self-drawn and did not infringe on any copyrights.
We hope that our revisions address all of your concerns and make our manuscript suitable for publication in your journal. We appreciate your time and dedication to our work and welcome any further questions or comments you may have.
Sincerely,
Lei Zhang